# Self-Compassion as a Mediator of the Relationship between Adult Women’s Attachment and Intuitive Eating

**DOI:** 10.3390/nu13093124

**Published:** 2021-09-07

**Authors:** Noémie Carbonneau, Mélynda Cantin, Kheana Barbeau, Geneviève Lavigne, Yvan Lussier

**Affiliations:** 1Department of Psychology, Université du Québec à Trois-Rivières, Trois-Rivières, QC G8Z 4M3, Canada; melynda.cantin@uqtr.ca (M.C.); genvievelavigne@hotmail.com (G.L.); yvan.lussier@uqtr.ca (Y.L.); 2School of Psychology, University of Ottawa, Ottawa, ON K1N 6N5, Canada; kbarb006@uottawa.ca

**Keywords:** adult attachment, self-compassion, intuitive eating

## Abstract

Despite growing interest in intuitive eating—a non-dieting approach to eating that is based on feeding the body in accordance with physiological and satiety cues—research on its determinants is scarce. The present study aimed to examine the associations between dimensions of adult attachment (i.e., anxiety and avoidance) and intuitive eating, and the mediating role of self-compassion in these relationships. The sample comprised 201 French-Canadian young adult women (M = 25.1, SD = 4.6). Participants completed self-report questionnaires through an online survey. Results of the structural equation model demonstrated that attachment-related anxiety and avoidance were negatively associated with intuitive eating, and these relationships were at least partially mediated by self-compassion. Findings suggest that women who have high levels of attachment anxiety or avoidance engage in less intuitive eating partly because they are less self-compassionate. Results highlight the importance of self-compassion in facilitating adaptive eating behaviors in adult women, especially if they have an insecure attachment style to romantic partners.

## 1. Introduction

Attachment representations cultivated by early life experiences have been recognized as important correlates of eating behaviors [1,2]. The association between attachment and eating behaviors is thought to be partly explained by interpersonal relationship dynamics (e.g., interpersonal conflict [3]) and affect regulation [2,4,5]. However, attachment research has primarily focused on adult attachment and disordered or unhealthy eating [1], creating a knowledge gap of how attachment relates to more adaptive eating behaviors, such as intuitive eating (e.g., eating according to satiety cues or physical needs [6]). Furthermore, despite the breadth of knowledge on the protective role of self-compassion on disordered eating through increased affect regulation [7], no research to date has examined self-compassion as an explanatory mechanism between adult attachment styles and intuitive eating.

Given that previous research demonstrates that self-compassion—the ability to be kind and caring toward oneself during times of distress—is associated with both attachment [4,5,8] and intuitive eating [9,10], the present research examined the mediating role of self-compassion in the relationships between two dimensions of attachment (i.e., abandonment anxiety and avoidance of intimacy) and intuitive eating. These relationships were examined in adult women because previous research indicates that women are less self-compassionate than men [11], suggesting that lowered self-compassion could potentially better explain the relationship between attachment anxiety and avoidance and less adaptive eating behaviors in women specifically. Men and women’s attachment styles have been shown to be associated with disordered eating through distinct emotion regulation difficulties [4], further suggesting that there may exist gender differences in coping strategies that mediate the relationship between attachment and eating behaviors. For these reasons, we sought to explore how young adult women’s attachment to others relate to their intuitive eating through their degree of self-compassion, an emotion-focused coping mechanism, to shed light on key interpersonal processes that are crucial for the development of adaptive eating behaviors in adult women.

### 1.1. Attachment Theory

Attachment theory [12,13] posits that humans are motivated to seek proximity to significant others (attachment figures) during times of need or distress due to an innate psychobiological system for survival (the attachment behavioral system [14]). Interactions with attachment figures, including their degree of availability and support, provide the foundation for mental representations of the self and others called “internal working models”. When attachment figures are sensitive and responsive, individuals form positive working models of the self (i.e., worthiness of care, the self is acceptable) and perceive a reliable (secure) source of support from others [14,15]. When attachment figures are inconsistently supportive or unresponsive, negative working models of the self are formed (i.e., not worthy of care, the self is not acceptable) because proximity seeking fails to provide comfort, and in turn, attachment figures (i.e., working model of others) are perceived as an unreliable (insecure) source of support [15]. Working models of others are thought to curate expectations about who will serve as attachment figures and how accessible and responsive they will be [16]. Individual differences in attachment styles reflect these interpersonal experiences with attachment figures [17] and are associated with distinct behavioral and affective patterns [2,13]. In the case of adult attachment, relationship-specific attachments continue to develop across the lifespan; therefore, an adults’ attachment style may be best represented by more recent compared to early life interpersonal experiences [18]. 

### 1.2. Attachment Styles

Adults’ attachment styles are conceptualized and measured based on two dimensions: attachment anxiety and avoidance [19,20]. An individual’s placement on these two dimensions reflects their degree of attachment security and the affect regulation strategies they employ when in distress with those scoring high on either dimension having more insecure attachments and using less adaptive emotion regulation strategies [14]. Attachment anxiety refers to an individual’s degree of worry about being underappreciated or abandoned by significant others [21]. Due to their preoccupations with abandonment, highly anxious individuals possess conflicting working models of others, such that they are hopeful but not confident that they can provide care and security [21]. A negative working model of the self is created from these concerns, and proximity to attachment figures is used to compensate for this sense of insecurity [22]. Preoccupation with abandonment and reassurance seeking behaviors are associated with greater relational distress and interpersonal conflict due to chronic anxiety about the stability of relationships with others [23,24]. Additionally, those with higher attachment anxiety tend to use emotion-focused/hyperactivating coping strategies when distressed, such as overexpressing or overexaggerating their feelings and fears, which sustain or escalate their concerns [25]. On the other hand, attachment avoidance refers to an individual’s degree of comfort with closeness and emotional intimacy in relationships [21]. Individuals who are avoidant have negative working models of others, such that they perceive others as being emotionally unavailable or conceive support seeking as undesirable [21]. For these reasons, they strive to maintain control and autonomy in their relationships [26]. Despite this view of others, those who are avoidant can have either positive (i.e., high self-esteem) or negative self-perceptions (self-critical [27]); however, their working model of others can propel them to employ distancing/deactivating coping strategies, such as emotional suppression, to maintain self-reliance and behavioral independence [14]. In turn, these distancing behaviors and negative working models of others are associated with lower relationship quality (i.e., connectedness, support) and satisfaction [24].

### 1.3. Adult Attachment and Eating Behaviors

Given that proximity seeking behavior to attachment figures is thought to function as an emotion regulation resource during times of distress, attachment styles have been examined as key contributors to mental illness and resilience [14]. Most notably, attachment theory has been used to understand how interpersonal processes, such as interpersonal style, affect regulation, and self-concept, underlie the development of eating disorders [2,28]. Reviews of the literature have suggested that attachment insecurity is more prevalent among individuals with eating disorders compared to healthy controls [2,28,29], further suggesting the strong role of interpersonal styles and conflict on eating patterns. According to Zachrisson and Skarderud [28], there are four interdependent mechanisms that explain the relationship between attachment styles and eating behaviors. First, insecure attachment is viewed as a vulnerability: those with insecure attachments tend to have worse mental and physical health, including poorer eating habits [28]. Second, insecure attachment is associated with emotion regulation difficulties: those with insecure attachments have been found to utilize more maladaptive coping strategies, including the use of food to nullify or escape negative affect [28]. Third, insecure attachment is associated with negative self-perceptions: those with insecure attachments tend to have negative working models of the self (i.e., view the self as worthless, being overly critical and perfectionistic [30]), which in turn can lead to feelings of shame and anxiety [28]. Experiences that induce shame and anxiety have been shown to be associated with forms of emotional eating, such as binge eating [31]. Finally, fourth, insecure attachment is associated with relational difficulties: those with insecure attachments have been shown to resort to hyperactivating or deactivating strategies to avoid relational distress, which prevents them from acquiring the social skills necessary to thrive and receive support in their relationship [28]. Additionally, these affect regulation strategies can increase interpersonal conflict [14], which in turn, increase negative affect and the use of secondary maladaptive coping strategies, such as disordered eating behaviors [32]. Although these mechanisms are discussed independently, poor self-perceptions and relationship difficulties are related to maladaptive coping strategies, including the use of food to compensate for affect dysregulation, and therefore have implications on eating behaviors directly and indirectly through emotion regulation deficits and poor stress management [1].

A meta-analysis examining the associations between attachment styles and eating behaviors more broadly found that individuals with insecure attachment styles (e.g., preoccupied, avoidant, fearful) were more likely to engage in unhealthy eating behaviors (e.g., binge eating, dieting, and unhealthy food consumption [1]). Additionally, higher attachment avoidance was associated with poor diet quality [33]. However, studies did not find a significant association between secure attachment and healthy eating behaviors (e.g., consumption of vegetables, high quality diet [33,34]). These findings suggest that attachment insecurity is associated with eating behaviors characterized by dysregulation, such as loss of control over eating, eating outside of satiety cues, emotional eating, food restriction, and lack of congruence with bodily needs (i.e., unhealthy food consumption). A more adaptive form of eating that encompasses both unconditional permission to eat (vs. restriction) and a more mindful approach to eating (vs. loss of control) is intuitive eating [6]. 

### 1.4. Intuitive Eating

Intuitive eating involves the consumption of food in accordance with internal satiety and hunger cues and thus physiological signals are relied on to regulate eating behavior [35]. However, intuitive eating also encompasses several other components that explain its positive relationship with health and wellbeing. According to Tylka and Kroon Van Diest [35], intuitive eating comprises four main elements: (1) unconditional permission to eat, including permission to eat any food that is desired (vs. having forbidden foods); (2) eating for physical rather than emotional reasons, such that eating is not in response to affective states; (3) reliance on internal hunger and satiety cues to determine when, what, and how much to eat; and (4) body–food choice congruence, which refers to choosing foods that serve dual purposes in terms of desire and nutrition to fulfill bodily needs. 

Considering that intuitive eating requires interoceptive awareness, does not impose dietary restriction for weight loss, and does not encourage the categorization of food to be “good” or “bad”, intuitive eating is negatively associated with food restriction and binge eating, shape and weight concerns, and internalization of the thin-ideal [36]. Intuitive eating is also positively associated with greater general wellbeing, positive self-perceptions and attitudes, such as self-esteem, self-compassion, and other resilient attributes, such as mindfulness [36]. Given the vast number of psychosocial correlates of intuitive eating, a theoretical model by Avalos and Tylka [37] has been proposed that explains the associations between early life environments, internal and external monitoring of the body, and intuitive eating. This model posits that intuitive eating is developed by positive interpersonal environments, specifically environments that are accepting and give unconditional positive regard. In turn, individuals exposed to these types of environments develop positive self-perceptions more generally and toward the body, while also feeling accepted by others. Accordingly, these individuals do not feel pressured to follow dieting regimes to obtain the thin-ideal, but instead, are encouraged to look inward and trust and appreciate their bodily functions and needs. Those who appreciate their bodies and are more attuned to their needs are thought to be more cognizant of and able to regulate their eating behaviors in accordance with physiological cues (i.e., hunger, satiety) compared to external factors. 

Even though intuitive eating is thought to be developed by interpersonal environments, very few studies have examined the relationship between attachment styles and intuitive eating. What is currently known is that individuals high in attachment avoidance engage in less intuitive eating [38]. This association was shown to be fully mediated by body appreciation [38]. In line with Avalos and Tylka’s theory of the development of intuitive eating [37], the link between interpersonal styles and intuitive eating is mediated by self-perceptions and how individuals treat themselves and their bodies. Self-compassion, a form of self-relating that is kind, accepting, and gives unconditional regard toward aspects of the self that are perceived to be inadequate, may also mediate the relationship between attachment and intuitive eating given that self-compassion is also cultivated by interpersonal environments [39,40,41] and is positively associated with intuitive eating [42]. 

### 1.5. Self-Compassion

Self-compassion has been identified as a resilience factor given its positive associations with a host of psychological and physical health outcomes [43,44]. In particular, self-compassion has been shown to be robustly negatively associated with negative body image and eating disorder symptomatology, primarily by improving affect regulation [7]. The affect-regulation model of self-compassion, also known as the self-regulation resource model of self-compassion [45], has been empirically tested and demonstrates that self-compassion replenishes resources for self-regulation by lowering negative affect and increasing positive affect, which partly explains why individuals higher in self-compassion engage in health-promoting behaviors, including healthy eating behaviors [45]. 

Self-compassion comprises three interacting components that aid in stress management by collectively mitigating negative affect when facing negative life experiences [46,47]. These three elements are: self-kindness, common humanity, and mindfulness [48]. First, self-kindness refers to showing kindness to oneself, being supportive and understanding of one’s own condition, and having a gentle and encouraging inner dialogue instead of a harsh and derogatory one. Second, common humanity implies recognizing that all human beings fail and are imperfect. Imperfection is part of the human condition; therefore, acknowledging that others are also imperfect protects us from feeling alone and isolated from others [49]. Finally, self-compassion implies awareness of one’s negative thoughts and emotions so that one can approach them with serenity and in a balanced way (i.e., mindfulness). Mindfulness of negative emotions and thoughts means that an individual does not “self-identify” with them and become caught up and overwhelmed [48]. Rather than mixing negative perceptions with themselves, individuals acknowledge that their thoughts and emotions are just that, which helps dispel their irrational beliefs [50]. Previous studies employing self-compassion interventions have demonstrated that increased self-compassionate coping is conducive to better emotion regulation as indicated by lower levels of negative affect [46] and reduced emotionally-driven eating as indicated by lower bulimic symptoms [51]. These findings may suggest that women’s degree of self-compassion may play a role in how they perceive and respond to negative interpersonal environments, and in turn, relate to how they approach eating. 

### 1.6. Self-Compassion and Adult Attachment

Similar to other self-attitudes, self-compassion is thought to be developed and facilitated through interpersonal relationships with key attachment figures [40,49,52,53]. For instance, young adults who show little self-compassion are more likely to have grown up with a critical mother and to have less secure attachment [40,41]. Conversely, individuals who have been raised in safe environments and who have experienced caring, loving, and authentic relationships with their attachment figures should develop and access self-compassion more easily [40,52]. More recently, there is evidence supporting this notion, such that compassionate parenting is shown to facilitate adolescents’ self-compassion [39]. Additionally, correlations between mothers and adult daughters’ self-compassion have been found [54] further suggesting that interpersonal environments that are warm and encourage compassionate dialogue can facilitate self-compassion in others, including adult women. 

Regarding insecure attachment styles, self-compassion has been shown to mediate the relationship between attachment anxiety and wellbeing [55,56], attachment anxiety and body appreciation [57], and attachment anxiety and psychological adjustment [58]. Individuals with high levels of attachment anxiety may be less self-compassionate due to their tendency to be harder on themselves, to self-identify with negative emotions (i.e., anxiety of partners abandoning them turns into feelings of abandonment and worthlessness), and exaggerate their distress [40,41,56]. In turn, these perspectives have negative implications on their self-concept (i.e., body image) and psychological wellbeing. However, the relationship between attachment avoidance and self-compassion is complex. Individuals with high levels of avoidance of intimacy can display both positive and negative self-views [17]; however, they are more likely to defensively improve their self-perceptions and ignore their vulnerabilities [59]. Thus, it is plausible that the avoidance of intimacy would be associated with less self-compassion because people with high levels of avoidance of intimacy have difficulties accepting their personal flaws and try to deny their shortcomings. However, some studies have not found an association between avoidance of intimacy and self-compassion [40,41], while others have found that self-compassion mediates the relationship between attachment avoidance and emotional distress and anxiety in clinical populations with mixed anxiety and depression [60]. 

### 1.7. The Interplay between Attachment, Self-Compassion, and Intuitive Eating

Although the relationship between attachment and intuitive eating has yet to be thoroughly investigated, self-compassion is shown to facilitate intuitive eating by improving self-perceptions, such as body appreciation and acceptance [9,10]. Additionally, one’s own self-compassion is shown to mediate the relationship between compassionate interpersonal behaviors and emotional eating [39] suggesting that self-compassion may explain the relationship between attachment style and intuitive eating. Furthermore, daily fluctuations in adult women’s self-compassion has shown to influence their degree of intuitive eating on any given day, demonstrating that self-compassionate coping in response to everyday challenges [9], including interpersonal conflicts, may be associated with women’s ability to engage in intuitive eating. It has been proposed that self-compassion is associated with a higher degree of intuitive eating through greater distress tolerance, such that a more soothing reaction to negative feelings may prevent emotionally-driven urges to under- or over-eat [9].

### 1.8. Current Study and Hypotheses

The objective of the current study is to address a prominent knowledge gap in eating behavior research by examining the associations between adult women’s attachment styles and intuitive eating through self-compassion. This area of research is especially important given that intuitive eating is associated many physical health benefits [61], offers a non-dieting approach to eating, and reduces disordered eating in the long-term [62]. Young female adults were targeted for this study because disordered eating mainly affects young adults and adolescents, and is more prevalent among women than men (e.g., [63,64]). In addition, young adult women are likely to be or soon become mothers; it is thus particularly important to study their eating behaviors as they are likely to pass them on to their children [54,65]. 

Based on current literature on attachment and self-compassion, it was hypothesized that (1) both dimensions of attachment insecurity (i.e., anxiety and avoidance) would be negatively associated with self-compassion. Due to the vast amount of literature that suggests that insecure attachment is associated with unhealthy and maladaptive eating behavior [1,2], it was hypothesized that (2) attachment anxiety and avoidance would be negatively associated with intuitive eating. Considering that self-compassion is shown to be positively associated with intuitive eating [9,10,42], it was hypothesized that (3) self-compassion would be positively associated with intuitive eating, and (4) will mediate the relationships between attachment anxiety and avoidance and intuitive eating.

## 2. Materials and Methods

### 2.1. Participants

The sample comprised 201 Canadian women between the age of 18–35 years (M = 25.1 years; SD = 4.6 years). Participants were recruited through Facebook advertisements. The present study is part of a larger project investigating young Canadian adults’ person-al characteristics and interpersonal relationships. The inclusion criteria for the larger pro-ject were being a young adult between the age of 18 and 35 years old and currently living in Canada. Only the female-identifying subsample is presented in this paper in line with the aims of the study. Digital consent was obtained prior to the commencement of the online survey. Participants were entered into a prize draw for one of three $50 CAD gift certificates.

The majority of women in the present sample were White (93%), were in a stable or fairly stable romantic relationship (74.1%), were cohabitating with their partner (69.9%), and reported a household income of less than $60,000CAD a year (52.2%). Most had completed a post-secondary degree (87.7%). Sociodemographic characteristics of the sample appear in Table 1.

### 2.2. Measures

*Adult attachment*. Adult attachment was evaluated with the 12-item Experiences in Close Relationships Questionnaire [66]. Six items assessed the degree of attachment avoidance (e.g., “I don’t feel comfortable opening up to romantic partners”), while the other six assessed the degree of attachment anxiety (e.g., “I need a lot of reassurance that I am loved by my partner”). Women were asked, on a 7-point Likert scale ranging from 1 (Strongly Disagree) to 7 (Strongly Agree), the extent to which each statement reflects their feelings or behaviors toward their romantic partners in general. After reverse scoring, items were averaged to create an overall score for each subscale. Structural validity of the French translated version of the scale was supported via exploratory factor analysis in two samples of French-Canadian adults [67]. In the present study, Cronbach alpha was 0.88 for each dimension. 

*Self-compassion*. Women’s degree of self-compassion was assessed using the Self-Compassion Scale-Short Form (SCS-SF [68]), which is composed of 12 items [69]. Women were asked to rate the extent to which each statement reflected how they treat themselves (e.g., “When I’m going through a very hard time, I give myself the caring and tenderness I need”) on a 5-point Likert scale ranging from 1 (Almost Never) to 5 (Almost Always). After reverse scoring, items were averaged to create an overall score. Previous research has provided evidence of the validity of the SCS-SF in college samples of women [68]. Scores on the SCS-SF in college samples had high internal reliability and nearly perfect correlation with the long form of the SCS (r = 0.97 [68]). In the present study, Cronbach alpha was 0.87.

*Intuitive eating*. Intuitive eating was assessed using the French version [70] of the Intuitive Eating Scale-2 [35]. This scale is composed of 23 items and evaluates four dimensions of intuitive eating: (1) eating for physical rather than emotional reasons, (2) reliance on hunger and satiety cues, (3) unconditional permission to eat, and (4) body and food choice congruence. Women were asked, on a scale 5-point Likert scale ranging from 1 (Strongly Disagree) to 5 (Strongly Agree), the extent to which they engage in intuitive eating (e.g., “I rely on my hunger signals to tell me when to eat”). After reverse scoring, items were averaged to create a global score of intuitive eating. Structural validity of this scale was supported by exploratory and confirmatory factor analysis in a sample of college men and women [35]. The French translation of this scale has been used in a previous study and demonstrated good internal reliability [70]. In the present study, Cronbach alpha was 0.91.

### 2.3. Data Analytical Plan

To determine if the variables of interest meet the assumptions for a mediation analysis, a series of linear regressions were conducted to determine if (1) attachment anxiety and avoidance was predictive of intuitive eating (c’ path; c1 and c2, respectively), (2) if self-compassion was predictive of intuitive eating (b’ path), and if attachment anxiety and avoidance were predictive of self-compassion (a’ path; a1 and a2, respectively). Specifically, the dependent variable (intuitive eating) is first regressed on the independent variables (attachment avoidance and anxiety). Then, the proposed mediator (self-compassion) is regressed on the independent variables (attachment avoidance and anxiety). To proceed, the relationships between the independent variables and the dependent variable as well as between the independent variables and the proposed mediator should be significant. When these requirements are met, a hierarchical linear regression model in which the dependent variable is first regressed on the independent variables in Block 1 and the proposed mediator is added in Block 2 can be tested. It is expected that, if mediations do exist, the direct relationships between the independent variables and the dependent variable will be weaker (partial mediation) or will even become non-significant (full mediation), in Block 2. Linear regression models were conducted using the IBM Corp. SPSS software.

For the main analysis, to examine the mediating role of self-compassion in the relationships between attachment anxiety and avoidance and intuitive eating, structural equation modeling analyses were conducted using the Amos statistical software [71]. Structural equation modeling offers the advantage of testing all proposed relationships (see Figure 1) simultaneously. The model tested in the present study included two exogenous variables (attachment anxiety and attachment avoidance) and two endogenous variables (self-compassion and intuitive eating). The maximum likelihood estimation method for analysis was used. Bias-corrected bootstrapping techniques (i.e., 1000 samples) were used to estimate the confidence intervals of the indirect effects, in which mediation is inferred if the confidence interval does not include 0 [72]. Specifically, we conducted bias-corrected bootstrapped 95% confidence interval estimates for the indirect effect of attachment anxiety on intuitive eating through self-compassion, and for the indirect effect of attachment avoidance on intuitive eating through self-compassion. Bootstrapping is a pre-specified resampling method used to generate the accuracy and range of estimates, including indirect effect distributions. This statistical technique resamples a dataset multiple times to generate a large number of simulated samples in order to estimate indirect effects and their 95% confidence interval. Past simulation studies have found that samples of over 200 participants were large enough to have high power and low Type I error rates when estimating indirect effects with this method (e.g., [63]). Standardized estimates are reported in the present study. 

## 3. Results

### 3.1. Preliminary Analyses

Inspection of missing data pattern revealed a non-significant Little’s MCAR test (χ^2^(df = 812) = 836.94, *p* = 0.265) which supports the null hypothesis that data are missing completely at random. It was thus decided that missing data would be handled through listwise deletion. The final sample comprised 201 participants. We further inspected, using chi-square statistics for categorical variables and univariate analysis of variance for continuous variables, whether the participants included in the final sample differed on a number of sociodemographic variables from those who were excluded. One significant difference emerged (χ^2^ (10) = 20.56, *p* = 0.024). Specifically, final participants were found to be significantly more educated than excluded participants.

Study variables were first inspected to determine if they met the basic assumptions for the proposed analyses. Inspection of the skewness and kurtosis values indicated that all variables had relatively normal distributions (highest skewness score was found for the attachment avoidance variable at 1.04). Furthermore, inspection of the scatterplot of residuals to predicted values showed no clear pattern, which supports the homoscedasticity assumption. Finally, the multicollinearity assumption was also supported as the highest correlation between the variables was between attachment anxiety and self-compassion at r = −0.43 and all variance inflation factors (VIF) were below 1.5 (highest VIF score was for self-compassion at VIF = 1.49).

Correlations between attachment avoidance, attachment anxiety, self-compassion, and intuitive eating are presented in Table 2. Results show that attachment anxiety and attachment avoidance are both significantly negatively correlated with self-compassion. Results further indicate that self-compassion is significantly positively correlated with intuitive eating. Finally, results demonstrate that attachment anxiety and attachment avoidance are both significantly negatively correlated with intuitive eating. 

A series of regressions were conducted to examine if self-compassion could function as a mediator between attachment and intuitive eating. First, a hierarchical linear regression was conducted with attachment variables and self-compassion as predictors of intuitive eating. Attachment anxiety and avoidance were entered into the first block and self-compassion was entered into the second block to examine if self-compassion predicted intuitive eating over and above both attachment dimensions. Results demonstrate that attachment anxiety and avoidance were significantly negatively associated with intuitive eating, (F(2, 198) = 10.569, *p* < 0.001, R^2^ = 9.6%; attachment anxiety: β = −0.18, *p* = 0.008; attachment avoidance: β = −0.24, *p* = < 0.001). When entering self-compassion into the second block, self-compassion explained an additional 3.1% of the variance in intuitive eating, (F(3, 197) = 9.591, *p* < 0.001). Results demonstrated that self-compassion was significantly positively associated with intuitive eating (β = 0.22, *p* = 0.009). When the influence of self-compassion was included, the significant negative relationship between attachment avoidance and intuitive eating held (β = −0.16, *p* = 0.027); however, attachment anxiety was no longer significantly associated with intuitive eating (β = −0.09, *p* = 0.224). These results suggest that self-compassion is associated with intuitive eating above and beyond adult attachment. Results of the hierarchical regression can be viewed in Table 3. Second, a linear regression was conducted between attachment anxiety and avoidance on self-compassion. Results demonstrated that both attachment anxiety (β = −0.42, *p* < 0.001) and avoidance (β = −0.38, *p* < 0.001) significantly negatively predicted self-compassion. These results suggest that self-compassion is a suitable mediator between both attachment dimensions and intuitive eating.

### 3.2. Mediation Model

The paths tested in the mediation model using structural equation modelling are displayed in Figure 1. Because all possible paths were specified in the model, it is said to be saturated. Saturated models have no degree of freedom; therefore, no fit indices are generated. Results of the structural equation modeling demonstrated that the direct relationship between attachment anxiety and intuitive eating was not significant (c_1_′: β = −0.09), whereas attachment avoidance was significantly negatively directly associated with intuitive eating, (c_2_′: β = −0.16). Additionally, both attachment anxiety and avoidance were significantly negatively associated with self-compassion (a_1_: β = −0.42; a_2_: β = −0.38). Furthermore, self-compassion was found to be significantly positively associated with intuitive eating (b: β = 0.22). Regarding the mediating role of self-compassion between attachment and intuitive eating, the indirect effect of attachment anxiety on intuitive eating through self-compassion and the indirect effect of attachment avoidance on intuitive eating through self-compassion were found both to be significant. Specifically, self-compassion was shown to significantly mediate the relationship between attachment anxiety and intuitive eating (95% CI = −0.175 to −0.027, *p* = 0.009) as well as the relationship between attachment avoidance and intuitive eating (95% CI = −0.154 to −0.025, *p* = 0.010). Thus, when all proposed relationships are modelled simultaneously in a structural equation model, results suggest that self-compassion acts as a significant mediator in the relationship between attachment styles and intuitive eating. Results further suggest that most of the variance associated with attachment anxiety in the prediction of intuitive eating is explained by levels of self-compassion as indicated by the non-significant c1′ path. However, some of the variance in attachment avoidance in the prediction of intuitive eating remains as the c_2_′ path is significant even though the contribution of self-compassion in the relationship is accounted for.

## 4. Discussion

The current study sought to examine if dimensions of insecure attachment conceptualized as high levels of anxiety and avoidance would be associated with lower intuitive eating in adult women. Further, we examined if self-compassion mediated the relationships between these attachment dimensions and intuitive eating. Our results were in support of our hypotheses such that women with higher attachment insecurity (i.e., higher attachment anxiety and avoidance) were less self-compassionate and engaged in lower intuitive eating. Furthermore, self-compassion was shown to fully mediate the relationship between attachment avoidance and intuitive eating and partly mediate the relationship between anxiety and intuitive eating. These results suggest that those with insecure attachments are less likely to engage in intuitive eating either fully or partly due to being less self-compassionate. 

Regarding the relationships between attachment and self-compassion, our results are in line with our hypotheses and previous research demonstrating that insecure attachment is associated with lower self-compassion [40,41,55,57,58]. Parents’ extension of compassion toward their children has been shown to facilitate their adolescent children’s self-compassion [39], suggesting that one’s ability to be self-compassionate is in part based on how attachment figures have treated them: the extension of compassion toward the self begins by others extending compassion toward you [55]. Although our study examined how adult attachment was associated with self-compassion, previous research has shown that the relationship between maternal attachment insecurity and self-compassion was mediated by peer and romantic insecure attachment [57]. These findings highlight that early life attachment experiences are influential on developing self-compassion by influencing types of attachments formed later in adulthood, which is congruent with the notion that the soothing system is developed by early attachment experiences [73]. Furthermore, the negative associations of attachment anxiety and avoidance with self-compassion is also explained by the tendency of highly anxious and avoidant individuals to develop negative working models of the self, which tends to prohibit them from treating themselves kindly and accepting their personal flaws with a mindful approach. Although attachment avoidance has not been consistently associated with negative self-perceptions, individuals high in attachment anxiety and avoidance have been found to be less soothing and colder toward the self, which partly explains why they experience lower wellbeing (i.e., higher depressive symptoms and lower quality of life [8]). 

Regarding the associations between attachment styles and intuitive eating, our results supported the hypotheses and were aligned with previous literature. Congruent with previous findings, more insecurely attached individuals tend to engage in less adaptive eating behaviors, such as lower intuitive eating [1,38]. In addition, when taking women’s level of self-compassion into consideration, attachment anxiety was no longer found to be directly associated with intuitive eating, which indicates that this association is fully mediated by self-compassion. Research examining attachment-eating behavior relationships in samples of women with eating disorders has shown that the relationship between attachment anxiety and disordered eating symptoms is mediated by the hyperactivation of emotions [32]. These findings may suggest that it is the affect regulation strategies that anxiously attached individuals employ that are related to their inability to self-regulate their eating behaviors and prohibit compensatory eating to cope with negative affect. 

In line with our hypotheses, self-compassion was shown to be associated with higher engagement in intuitive eating and mediated the relationships between insecure attachment and intuitive eating. There is growing literature to suggest that self-compassion not only protects women from engaging in disordered eating [7] but is also associated with the engagement in health-promoting behaviors [74], including multiple facets of adaptive eating behaviors (e.g., intuitive eating, healthy eating [42,75]). Self-compassion can be viewed as an individual-level factor (i.e., personal characteristic) and a within-persons factor (i.e., state), suggesting that it is a positive self-attitude that is developed from early life experiences (i.e., trait) and a coping strategy that can be used to deal with negative experiences or emotions (i.e., state [9]). From this perspective, self-compassionate individuals may be more prone to engage in intuitive eating because they are more likely to reject societal messages (e.g., stereotypes about ideal body types) and pressures to be thin due to their self-accepting view. In turn, this self-accepting perspective may protect them from engaging in dieting behaviors (i.e., food restriction, enrolling in dieting programs that label foods as “good or bad”) and emotional eating [76,77]. Furthermore, engaging in self-compassion provides benefits on affect regulation, which allows individuals to tolerate distressing feelings and situations [45]. This increased emotional tolerance may increase self-regulation capacity and reduce the likelihood of dysregulated eating, such as emotional eating and binge eating, especially in reaction to negative affect. In line with previous findings in other domains of wellbeing [55,57,58], self-compassion mediated the relationships between attachment and intuitive eating. These findings further our knowledge of the affect regulation mechanisms that explain the negative relationship between insecure attachment and adaptive eating behaviors. They also highlight the potential benefits of increasing self-compassion for promoting more adaptive eating behaviors in those who have high levels of attachment anxiety or avoidance, suggesting that self-compassion interventions may have a dual purpose in these individuals by both decreasing disordered eating and increasing intuitive eating. Further research is required to determine if self-compassion interventions can facilitate more adaptive eating behaviors while also reducing disordered eating behaviors. 

### Strengths, Limitations, and Future Directions

The present study has a number of significant strengths. It is among the first to examine the associations between attachment, self-compassion and intuitive eating, which fills a current knowledge gap in the literature on women’s development of adaptive, healthy eating behaviors. Furthermore, this study provides additional knowledge of the mechanisms underlying the relationship between insecure attachment and eating behaviors. This additional knowledge is particularly relevant to attachment theories of eating disorders, which often focus on the affect regulation pathways, including strategies that influence the maintenance of disordered eating in clinical populations [2]. 

Despite these strengths, the current study also has limitations. First, causal relationships between attachment, self-compassion, and intuitive eating cannot be inferred due to the correlational nature of the study design. Future research should examine the longitudinal associations between these variables to determine if self-compassion functions as an affective regulation strategy that explains the relationship between attachment and wellbeing outcomes or if self-compassion is a resilient trait that could protect individuals from developing an insecure attachment to others. Given that previous research has shown that early life attachments (i.e., parents) influence attachments to others later in life (i.e., peers, romantic partners [57]), it is plausible that self-compassion could function as a mediator between early life attachment and later life attachment. For instance, previous research has found that self-compassion and compassion for others mediate the relationship between insecure attachment and relationship quality [78]. These findings may suggest that self-compassion and the extension of self-compassion toward others has implications on communication styles with others, influencing relationship quality and perhaps attachment with others. Second, the present study used self-report measures, which may have led to socially desirable responding. Third, this study only examined attachment to romantic partners. Future research should examine how attachment to others influences women’s eating behaviors given that previous research demonstrates their unique and intersecting impact [57]. Finally, the sample was highly homogenous with most participants being White, young, and educated. This homogeneity of the sample limits the generalizability of the findings to a broader population. Future research should attempt to replicate the present findings with a more diverse sample.

## 5. Conclusions

Overall, our results demonstrated that higher levels of attachment anxiety and avoidance are associated with lower intuitive eating through lower self-compassion. These results suggest that individuals with higher attachment anxiety and avoidance possess less self-compassion, which fully or partly explains why they engage in less adaptive eating behaviors, such as intuitive eating. Results highlight the benefits of self-compassion for adaptive eating behaviors and may suggest that increasing self-compassion in adult women could support their ability to engage in intuitive eating. There exist many methods of increasing self-compassion in women that are both effective and cost-efficient, such as writing activities, that aid emotion regulation and lower disordered eating [46,51]. It is plausible that increasing women’s self-compassion through an intervention could help them cope with negative interpersonal experiences elicited by perceived abandonment (i.e., attachment anxiety) or lack of support (i.e., attachment avoidance), and in turn, help nurture their intuitive eating.

## Figures and Tables

**Figure 1 nutrients-13-03124-f001:**
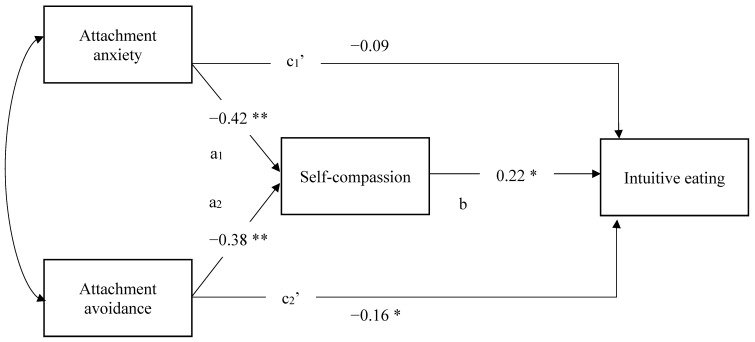
Structural Equation Model demonstrating the direct and indirect paths between attachment and intuitive eating. N = 201; * *p* < 0.05, ** *p* < 0.01. Curved arrow represents variables that are allowed to covary; solid lines represent direct effects.

**Table 1 nutrients-13-03124-t001:** Sociodemographic characteristics of the sample.

	n	%
Education (completed)		
-High school	24	11.9%
-College/CEGEP degree	91	45.3%
-University degree	85	42.4%
-Other	1	0.5%
Student status		
-No	96	47.8%
-Yes, part-time	21	10.4%
-Yes, full-time	84	41.8%
Employment status		
-No	41	20.4%
-Yes	160	79.6%
Relationship status		
-Single and not dating	38	18.9%
-Dating but not in a relationship	11	5.5%
-In a stable or relatively stable relationship	149	74.1%
-In an unstable relationship	3	1.5%
Cohabiting with partner		
-No	49	30.1%
-Yes	114	69.9%
Marital status		
-Not married	135	89.4%
-Married	16	10.6%
Children		
-No	159	79.1%
-Yes	42	20.9%

**Table 2 nutrients-13-03124-t002:** Descriptive statistics and bivariate Pearson correlations among variables in the model.

	Mean	SD	1	2	3	4
1. Attachment anxiety	4.2	1.6	-	0.05	−0.43 **	−0.19 **
2. Attachment avoidance	2.4	1.2		-	−0.40 **	−0.25 **
3. Self-compassion	2.8	0.7			-	0.32 **
4. Intuitive eating	3.4	0.7				-

Note. N = 201; ** *p* < 0.01.

**Table 3 nutrients-13-03124-t003:** Results of the hierarchical linear regression for intuitive eating.

Block		*β*	*p*
1	Abandonment anxiety	−0.18	0.008
	Avoidance of intimacy	−0.24	0.000
2	Abandonment anxiety	−0.09	0.224
	Avoidance of intimacy	−0.16	0.027
	Self-compassion	0.22	0.009

Note. N = 201. β = Standardized regression weight; *p* = *p*-value of the regression weight. Dependent variable: Intuitive eating. Block 1: regression of the dependent variable on the independent variables; Block 2: addition of the proposed mediator in the linear regression model.

## Data Availability

The data are available on request to the corresponding author.

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
