# Peer review of "Self-Compassion as a Mediator of the Relationship between Adult Women’s Attachment and Intuitive Eating"

_nutrients, 2021, doi:10.3390/nu13093124_

Round 1
Reviewer 1 Report
This is an excellent paper, I thoroughly enjoyed reading and reviewing it. It is rare that I read a paper where an introduction and a discussion section are as well written and as carefully laid out, and as compelling, as they are in this paper. Given the complexity of the topic, this is not an easy feat. I would encourage the authors to give the same level of detail in the writing of the methods and the results section of this paper, so that the impact of the work can be more fully appreciated by readers who are not experts in the statistical methods that are being used in this paper. Some suggestions on ideas for parts of the text that could specifically be expanded or further explained are outlined below. I would welcome the opportunity to read a revised version of this interesting and important paper.
In the methods section, it will be important to check that units are always shown when Data is shown. For example, in the first sentence of the results, the unit of years needs to be shown with the age.
In the methods section, I would liked to have understood more about the inclusion and exclusion criteria for the study.
In the methods section, I would like to have understood more about the population that was targeted for recruitment and why.
The abbreviation amos in the methods section is uncommon, and it would be better to use the full version rather than this abbreviated acronym.
Section 2.3 would benefit from being greatly expanded so that the statistical techniques are more clearly understandable by the types of readers who are interested in this field, but not necessarily knowledgeable in these specific statistical techniques. More explanation of, for example, assumptions underlying the methods, as well as the methods themselves, would be of benefit for this paper.
Table 2 needs more information, notably in a footnote, so that it is standalone understandable. For example, what statistical methods were used to calculate correlations, etc.
Table 3 also requires more information so that it is stand alone understandable, without reference to the text.
The results section is reasonably clear, but I need to read between the lines to try to understand the results. I think that the results section would benefit by being expanded with more words to explain these interesting analyses to readers that are not familiar with these statistical techniques. Alternately, and additionally, statistical review by somebody familiar with these statistical techniques would seem important.
Section 5, which is the conclusion, contains information and literature citations that I would have expected to have seen earlier in the text, where information about self-compassion interventions was first mentioned in the discussion. These literature citations should not be in the conclusion in my view, they should be incorporated earlier into the discussion. On the other hand, the conclusion needs to also Tone down the suggestion that interventions to increase self-compassion might enhance eating behaviors, given that, as the authors have acknowledged, this is a correlation study.
Reviewer 2 Report
Abstract: Very clear and well written.
Introduction: This is the most well written introduction I have reviewed in some time! It explains each component of the study (self-compassion, adult attachment, intuitive eating) in depth that is appropriate for the study and thouroughly reviews the existing literature of how these components are related, why it is important, and why the current study "matters". The hypothesis are listed clearly and supported by research. The only thing I would suggest editing would be to use more "couching" language when discussing attachment styles and outcomes: for example, instead of "those with insecure attachment styles are more anxious...etc." I suggest saying "tend to be" or something similar. Instead of "those with secure attachment styles should" I would suggest saying "it is theorized that those with secure attachment styles would develop....etc." or related.
Method:
Participants: Great table! What were the participant requirements; ie, what did the study advertise for and were there any variables that would make someone not eligible for the study? Were participants compensated in anyway? I recommend including this information.
Measures: well explained.
Data: how were missing data handled? Please include this information.
Results: Line 386 --> R2 instead of R2, please edit. Please add the model fit statistics (Chi2, CFI, RMSEA, etc.).
Overall, I think this paper will be well received and an important addition to the ED and related fields. This paper is well very well written. Thanks!
Reviewer 3 Report
nutrients-1350766
Self-compassion as a mediator of the relationship between adult women’s attachment and intuitive eating.
The current study examined whether self-compassion mediated the link between women’s avoidant and anxious attachment and their intuitive eating. The authors found support for this model, such that women with high anxious or avoidant attachment engage in less intuitive eating, particularly when they are lower on self-compassion. This suggests the importance of fostering self-compassion in helping women improve intuitive eating, especially among women with insecure attachment styles.
Overall, this was a well-written paper. Analyses were appropriate. Interpretations of findings were appropriate for the analyses. I have a few concerns that may improve the paper, below.
Introduction:
The introduction is 15 ¶ long, which is on the long side. It feels as though it could be successfully condensed somewhat. Personally, I attempt to aim for approximately 9-10¶ when writing my own introductions for full-length manuscripts (at least when there are word/page limits that I need to be mindful of).
Also the structure of the introduction was a little different from how I had been trained to write an academic paper (and from most I read and review). Typically, I follow the infundibuliform structure suggested by Bem (2003)[1] that the introduction starts off broad and becomes more narrow/focused as you work towards the “purpose of the current study” paragraph. The first two paragraphs in this introduction already seemed to be the “purpose of the current study” paragraphs, which was then repeated again, where expected, at the end of the introduction. This is well-written, however, so I view that more as a stylistic choice than a requirement (from my perspective) for publication.
Methods:
Was the sample size sufficient for mediation? It would be helpful to provide data on this. There are some published papers which report sample sizes needed for specific sizes of the indirect and direct effects for .8 power. It would be helpful to add this information.
The authors did not report on whether the variables met preconditions for regression (e.g., normality assumptions, homoscedasticity, non-multicollinearity)
The authors also did not report on proportion of missing data, whether it was missing at random, and what was done with missing data.
Results:
Results were reported appropriately and clearly. It would be helpful to clearly state “the indirect effect of avoidant attachment on intuitive eating, via self-compassion was X, and the direct effect was Y and the total effect was Z” or something similar, so that it was crystal clear for readers.
Discussion:
p. 10, lines 439-440: The authors use “one” and then “them” in the same sentence. I am aware that using they/them/their as singular gender neutral pronouns is more commonly accepted, but when not referring to a particular individual who prefers those pronouns, it still feels grammatically questionable for those of us who are old and have difficulty straying from our grammatical traditions. The authors may wish to reword. Similarly, is there a way to reword the sentence without the second person?
p. 11, line 462: The word order seems a little off. Perhaps reword to “when taking women’s level of self-compassion into consideration”
p. 11, line 482: Is there a way to reword this sentence that does not make it seem as though the authors believe that not being thin or fitting a sociocultural appearance ideal is a “personal flaw” (e.g., “perceived personal flaw”)?
p . 12, line 526: Grammar: “how attachment to others influence women’s”
The sample characteristics reported in Table 1 reflect a very “WEIRD” sample: Disproportionately White and educated (and relatively young as well). This should be noted as an additional limitation to the current study, given that it may not generalize to broader populations.
p. 12, lines 537-543: The authors make a leap to suggest that improving self-compassion would improve not only disordered eating, but also perceived abandonment or lack of support. Given the authors tested the reverse direction (but acknowledged that the cross-sectional data mean that this reverse situation could be more appropriate), this seems like a jump for the final conclusion sentence. It seems that focusing on interventions to address eating behaviors makes more sense with the hypothesized direction of influence and results presented (since anxious and avoidant attachment were not examined as DVs with self-compassion as an IV).
[1] In Darley, J. M., Zanna, M. P., & Roediger III, H. L. (Eds) (2003). The Complete Academic: A Practical Guide for the Beginning Social Scientist, 2nd Edition. Washington, DC: American Psychological Association.
